# The Relationship between Mercury Exposure Indices and Dietary Intake of Fish and Shellfish in Women of Childbearing Age

**DOI:** 10.3390/ijerph17134907

**Published:** 2020-07-07

**Authors:** Jeong-wook Seo, Byoung-gwon Kim, Young-seoub Hong

**Affiliations:** 1Environmental Health Center, Dong-A University, Busan 49201, Korea; jwseo@dau.ac.kr (J.-w.S.); medikim@dau.ac.kr (B.-g.K.); 2Department of Preventive Medicine, Dong-A University, Busan 49201, Korea

**Keywords:** total blood mercury, blood methyl mercury, hair mercury, urine mercury, women of childbearing age

## Abstract

Women of childbearing age who are susceptible to mercury exposure were studied to understand the relation between mercury intake through fish and shellfish consumption and mercury exposure indices from blood, hair, and urine samples. A total of 711 women of childbearing age from coastal areas with a high concentration of mercury exposure in Korea were studied. Data were collected on demographic characteristics, dietary intake of fish and shellfish using the simple Food Frequency Questionnaire. Mercury concentration was estimated from the collected samples of blood, hair, and urine. The geometric mean of blood methyl mercury concentration of mercury exposure through seafood was 3.06 μg/L for the low tertile, 3.12 μg/L for the middle tertile, and 3.60 μg/L for the high tertile, indicating a clear tendency of blood methyl mercury to increase as the mercury exposure by fish and shellfish intake ascended. For total blood mercury and hair mercury, the middle and high tertiles had higher values than the low. Mercury exposure through fish and shellfish intake is a main factor for an increase of blood methyl mercury concentration in women of childbearing age. More attention needs to be paid to mercury exposure through seafood intake, considering the serious effect mercury concentration has on women of childbearing age.

## 1. Introduction

Mercury (Hg) toxicity varies depending on many factors including the type of chemical and the way it infiltrates the system, the amount and duration of exposure, and one’s susceptibility. Particularly in pregnant women, elemental Hg (E-Hg) and inorganic Hg (I-Hg) might pass through the blood–placenta barrier [1] and cause hereditary toxicity to the fetus [2]. High-dose exposure to I-Hg in a fetus can result in a comprehensive range of developmental disorders such as mental retardation, cerebellar ataxia, dysarthria, limb deformities, altered physical growth, sensory impairments, and cerebral palsy [3]. Given that a fetus is more sensitive to toxins relative to children or adults and blood methyl Hg (MeHg) in a fetus tends to be higher than that in the childbearing mother [4], there could be a heightened risk for miscarriage and fetal deformity when the mother is exposed to the toxic compound, even though the amount of exposure was below the permitted level with no indication of poisoning [5]. It is therefore necessary for women of childbearing age to be cautious, as all have the potential to become child bearers.

Hg exposure in women of childbearing age has various clinical indications. Exposure to MeHg is understood to be mainly due to consumption of seafood, causing biological magnification of Hg in the system [6]. The Korea Ministry of Food and Drug Safety(KFDS) carries out risk assessment of Hg and MeHg of fish and shellfish [7], and related studies on specific coastal areas have been conducted [8,9].

MeHg compounds in an organism move around via red blood cells, and therefore are easily dispersed to different tissues in the body, even permeating through the blood–brain barrier, and affecting the central nervous system [10]. In general, E-Hg is most likely to be absorbed by inhaling oxygenated Hg vapor [11]. Acute exposure to highly concentrated Hg vapor can affect the respiratory system, and neurological symptoms can occur in cases of chronic exposure [12]. As for I-Hg, one is most likely to be exposed orally, but the absorption rate is known to be low [13]. One of the most serious health-threatening effects of I-Hg is kidney dysfunction caused by Hg concentration in the organ [14]. Blood, hair, and urine samples are typically used to measure Hg exposure concentration because direct examination of each affected tissue is not always practically viable, even though it might be the best method to gauge the exposure level. Blood can reveal a relatively recent Hg exposure history [15]. Total blood mercury (T-Hg) includes I-Hg and organic Hg (O-Hg), with I-Hg having been reported to account for approximately a third of T-Hg [16]. However, the reported ratio of MeHg in blood T-Hg differs widely, from lower than 30% [17,18,19] to higher than 90% in a study conducted with a rural fishing community [20,21]. These studies share a tendency that the MeHg concentration and ratio were drastically raised as the dietary intake of fish and shellfish increased. Given that MeHg accumulates in the hair [22,23], this is a very useful indicator for assessing long-term MeHg exposure [24]. Urine, on the other hand, is often used to estimate relatively chronic I-Hg exposure [25]. A continuous research effort has been ongoing outside of Korea, with the focus on Hg exposure and its effect on general public health. Relevant guidelines also have been made available for relatively vulnerable groups, such as children and women of a childbearing age, as well as the general population. In contrast, Korea has failed to provide guidelines and care for those who are at particularly high risk; however, general mercury exposure and exposure due to occupation have been well studied. The present study estimated T-Hg concentration based on biomarkers (including blood MeHg) collected from blood, hair, and urine specimens from a group of women of childbearing age from geographical regions where seafood consumption is particularly high. In addition, we aimed to relate Hg exposure through seafood consumption to Hg concentration and other biomarkers, in order to provide basic data for standards and guidelines to be established in future.

## 2. Materials and Methods

### 2.1. Study Design and Data Collection

Women of childbearing age, 19–49 years, were recruited between June and October 2010 from the coastal areas of Busan, Ulsan, and Kyongnam province in Korea for study participation. In the process of sampling the study population, we divided these areas into 28 units, for which 25 participants were sampled from each unit. Each geographical unit contained equal proportions of those in their 20s, 30s, and 40s so that demographic characteristics would be fully considered. Blood, hair, and urine specimens were collected from each participant to analyze blood T-Hg (*n* = 711), blood MeHg (*n* = 90), hair Hg (*n* = 359), and creatinine-adjusted urine Hg (*n* = 616).

### 2.2. Questionnaire Survey

A questionnaire survey was conducted by trained researchers in a one-on-one format. The survey collected data on gender, age, economic status, general educational characteristics, drinking habits, and exposure to Hg via amalgam treatment. Separately, a 24-h dietary recall (24HR) and simple Food Frequency Questionnaire (simple FFQ) were used to investigate dietary intake. Intake of items from 16 different food groups including seafood was probed by 24HR, after which resultant Hg intake was estimated through Computer-Aided Nutritional Analysis program (Can-pro ver3.0, Korea Nutrient Study Society, Seoul, Korea). Simple FFQ estimated the amount of Hg intake that occurred from consuming 22 different seafood items including 20 regular fish and shellfish as suggested by the Korea National Institute of Food and Drug Safety Evaluation (KNIFDSE) and two large fish (sharks and whales) that were understood to have high concentrations of Hg. The estimation was operationally defined as “Hg exposure through fish and shellfish.” The Hg concentration in the 22 seafood items were taken from KFDS and data from Kim et al. [26]. The defined variable was divided into low, middle, and high tertiles.

### 2.3. Analysis of Blood, Hair, and Urine Mercury Concentration

In order to prevent coagulation in the blood samples, EDTA-treated 3-mL vacutainers (Beckton and Dickton, Franklin Lakes, NJ, USA) were used to obtain venous blood. Hair specimens were collected from the occipital area of the head and each strand was taken as closely to the scalp as possible. Any contamination (from dye or hair perm) was checked. Spot urine was utilized to take urine samples. A Direct Mercury Analyzer (DMA80, Milestone Co, Fort Collins, CO, USA) was used for the analysis of total blood, hair, and urine Hg. To acquire a calibration curve, we prepared four reference samples including total blood and urine (Hg 2–8 ppb) and hair (Hg 5–10,000 ppb) by diluting standard solution (Waco Co, Minato, Tokyo, Japan). Re-analysis was carried out when the linearity (r) of the calibration curve was below 0.998. The limit of detection (LOD) was estimated at 0.0021 ng. A methyl mercury analyzer (MERX, Brooks Rand, Seattle, WA, USA) was used to analyze blood MeHg. For a calibration curve, we produced six reference samples of 1–250 pg/mL by using a methyl mercury standard stock solution (Brooks Rand, Seattle, WA, USA). A calibration curve was drawn up by the same method, and the LOD was 0.001 ng/L. The analytic method of gold amalgamation was used for total blood, hair, and urine Hg, with reference to the method 7473 of EPA [27]. The estimation of blood MeHg was based on the EPA method 1630 [28]. The calibration curve was checked in all 40 samples using the reference samples. Commonly used standard agents were used for internal quality certification and management: Whole Blood metals control level 1, 2 (Seronorm, SERO AS, Billingstad, Norway) for blood T-Hg, Toxic Metals in Caprine Blood 995c level 3 (National Institute of Standards and Technology, Gaithersburg, MD, USA) for blood MeHg, MESS-3 (NRC; National Research Council Canada, Montreal Rd, Ottawa, ON, Canada) for hair Hg, and Lyphochek Urine Metals Control level 2 (Bio-Rad Laboratories, Hercules, CA, USA) for urine Hg. As a proof of external quality approval and management, certifications for total blood Hg and urine Hg were secured from the German External Quality Assessment Scheme (G-EQUAS) of Friedrich-Alexander University. Accredited by the Korean Occupational Health and Safety Agency (KOSHA), the present institute published a paper on the estimation of MeHg in human blood by GC-CVAFS [26].

### 2.4. Statistical Analysis

SAS (Version 9.4, SAS Institute, Cary, NC, USA) was used for statistical analysis. With blood, hair, and urine Hg concentration data, a log transform was performed on the skewed distributions (skewness > 0) before the analysis. Independent samples *t*-tests and ANOVA were conducted to examine the differences between participants in terms of their individual characteristics, while Hg concentration was calculated after adjusting for age, educational status, monthly income, drinking habits, and amalgam treatment history, with an aim to measure Hg intake through fish and shellfish. The significance level was set at below 5% for all statistics.

### 2.5. Ethics Statement

A data collection agreement (for the questionnaire and blood, hair, and urine samples) was secured from all participants after providing them with a full explanation of the purpose and procedure of the study. We then arranged for the personal information collected and the results of specimen analysis to be available to the participants. The present study received approvals from the Korea National Institute of Environmental Research and the Institutional Review Board (IRB No. EHRD-220-2010.8.11).

## 3. Results

### 3.1. Mercury Intake through Fish and Shellfish

When 711 participants were surveyed using the 24HR, the mean proportion of Hg intake through seafood consumption to total Hg intake through 16 different food groups [± standard deviation] was 35.8% [± 31.4%]. The proportion by quartile according to the total Hg intake increased from the first quartile (6.18% [± 10.43%]) to the fourth (71.52% [± 18.20%]). Namely, as the total Hg intake increased, the proportion also tended to increased. The correlation coefficient between total Hg intake and proportion of Hg intake from fish and shellfish was 0.674 (*p* < 0.001) (Figure 1).

### 3.2. Total Blood, Blood Methyl, Hair, and Urine Mercury Concentration

The blood T-Hg concentration for the 711 participants was analyzed to obtain a geometric mean (95% confidence interval) GM (95% CI) of 4.11 (3.90–4.33) µg/L. Blood T-Hg concentration significantly increased with age (3.69 [3.49–3.90] µg/L for those aged 19–29 years, 4.08 [3.61–4.60] µg/L for those aged 30–39 years, and 4.65 [4.36–4.96] µg/L for those aged 40–49 years, *p* = 0.003). There was a statistically significant difference for educational status (high school or lower: 4.51 [4.20–4.84] µg/L, university student: 3.47 [3.00–4.01] µg/L, university graduate: 4.51 [4.20–4.84] µg/L, *p* < 0.001) and amalgam treatment history (present: 4.30 [4.12–4.48] µg/L, absent: 3.82 [3.38–4.30] µg/L, *p* = 0.031). When the distribution of Hg exposure through fish and shellfish was divided into tertiles, the middle and high tertiles had higher values than the low tertile (low 3.84 [3.62–4.07] µg/L, middle 4.31 [4.07–4.55] µg/L, high 4.19 [3.66–4.80] µg/L), but with no statistical significance (*p* = 0.307). The tendency of GM of blood T-Hg as adjusted for demographic and lifestyle characteristics was found to stay constant. Data from 90 participants were analyzed for their blood MeHg to give a GM (95% CI) of 3.32 (3.03–3.64) µg/L. Compared to the 19–29-year age group (2.96 [2.63–3.33] µg/L), the concentration was higher in the 30–39- (3.59 [3.06–4.20] µg/L) and 40–49-year (3.43 [2.81–4.20] µg/L) age groups, but with no linear tendency nor statistical significance (*p* = 0.206). No significant differences were found in the concentration by educational status, monthly income, drinking habits, or amalgam treatment history. Although the high tertile showed a sizeable increase in blood MeHg induced by Hg exposure through fish and shellfish in comparison to the middle and low tertiles (low 3.20 [2.70–3.80] µg/L, middle 3.21 [2.76–3.74] µg/L, high 3.58 [3.02–4.25] µg/L), there was no statistically significant difference (*p* = 0.553). When demographic and lifestyle characteristics were adjusted for, blood MeHg concentration showed a clear tendency to increase as Hg exposure through fish and shellfish accumulated (low: 3.06 [2.48–3.77] µg/L—middle: 3.12 [2.59–3.76] µg/L, high: 3.60 [2.95–4.40] µg/L, *p* = 0.394). Hair Hg concentration data were analyzed for 359 participants to give a geometric mean (95% CI) of 0.97 (0.92–1.03) µg/g. Relative to the younger age group (19–29 years: 0.84 [0.77–0.93] µg/g), older groups (30–39 years: 1.06 [0.97–1.16] and 40–49 years: 1.08 [0.94–1.23] µg/g) were found to have higher concentration of Hg and the difference was statistically significant (*p* = 0.001). No significant difference was found in hair Hg concentrations by educational status, monthly income, and drinking habits, with no distinctive tendency to report. Those with a history of amalgam treatment had statistically higher concentration (1.03 [0.96–1.11] µg/g), relative to those with no such history (0.90 [0.83–0.97] µg/g) (*p* = 0.014). The tendency of the GM of hair Hg as adjusted for demographic and lifestyle characteristics found to stay constant. The GM by concentration of Hg exposure through fish and shellfish was higher in the middle and high tertiles than the low tertile but with no statistical significance (low: 0.94 [0.85–1.03] µg/g, middle: 1.00 [0.91–1.10] µg/g, high: 0.99 [0.90–1.08] µg/g, *p* = 0.583). The same held true for the adjusted GM (low: 0.89 [0.80–0.99] µg/g, middle: 0.93 [0.84–1.03] µg/g, high: 0.92 [0.83–1.03] µg/g, *p* = 0.739). Urine Hg was analyzed in 616 participants to obtain a GM (95% CI) of 3.32 (3.03–3.64) µg/g-cr. When the GM was adjusted for age, urine Hg was found to increase with age (1.45 [1.24–1.71] µg/g-cr for those aged 19–29 years, 1.74 [1.53–1.98] µg/g-cr for those aged 30–39 years, and 1.90 [1.65–2.19] µg/g-cr for those aged 40–49 years) with statistical significance (*p* = 0.028). In terms of monthly income, the value was highest for KRW 1.99 million or less group (1.98 [1.71–2.30] µg/g-cr) and lowest for the KRW 3.00–3.99 million group (1.43 [1.21–1.68] µg/g-cr), with a statistically significant difference (*p* = 0.018) but no linear tendency. There were no statistically significant differences in urine Hg by educational status, monthly income, drinking habits, and amalgam treatment history. No correlation was found between urine Hg concentration and Hg exposure through fish and shellfish (low: 1.67 [1.50–1.85] µg/g-cr, middle: 1.92 [1.74–2.11] µg/g-cr, high: 1.60 [1.37–1.87] µg/g-cr, *p* = 0.553), and the same result held for the adjusted GM (low: 1.67 [1.45–1.93] µg/g-cr, middle: 1.86 [1.63–2.14], high: 1.54 [1.34–1.78] µg/g-cr, *p* = 0.553) (Table 1).

### 3.3. Correlations among Mercury Concentrations

Pearson’s correlation coefficients between blood T-Hg and blood MeHg, hair Hg, and urine Hg were 0.886 (*p* < 0.001), 0.461 (*p* < 0.001), and 0.211 (*p* < 0.001), respectively. Blood T-Hg showed a very strong positive correlation with blood MeHg, while showing intermediate concentration of positive correlation with hair Hg and week positive correlations with urine Hg. A positive correlation was found between blood MeHg and hair Hg (r = 0.600, *p* < 0.001) (Figure 2).

### 3.4. Proportion of Blood MeHg/T-Hg Concentrations

The arithmetic mean (AM) (95% CI) of ratios of MeHg against T-Hg in 90 participants whose MeHg concentration was analyzed was 76.51% [73.62–79.39%]. The 40–49-year age group had the lowest value at 74.10% [69.36–78.84%], whereas the 30–39-year age group had the highest value at 78.07% [72.45–83.69%]. There was no statistically significant tendency related to the age variable, nor to educational status, monthly income, drinking habits, or amalgam treatment history. The ratio of the concentration of Hg exposure through fish and shellfish was higher in the high tertile than in the low and middle tertiles, but with no statistically significant difference (low: 75.70% [70.88–80.52%], middle: 75.64% [69.77–81.51%], high: 77.70% [72.87–82.54%], *p* = 0.793). The same was true for the adjusted ratio (low: 76.30% [69.96–82.63%], middle: 77.54% [71.53–83.55%], high: 81.24% [75.56–86.91%], *p* = 0.386) (Table 2).

### 3.5. Proportions of Total Blood Mercury Concentrations Exceeding the Standard Criteria

For T-Hg, the AM (95% CI) of the ratios of those who surpassed the HBM-I standard threshold of 5.0 µg/L was 34.18% (30.68–37.67%). The ratios were found to significantly increase with age (21.03% [15.76–26.30%] for those aged 19–29 years, 37.50% [31.62–43.38%] for those aged 30–39 years, and 44.39% [37.68–51.10%] for those aged 40–49 years, *p* < 0.001), while there were statistically significant differences by educational status (high school degree or lower: 40.66% [33.46–47.86%], university students: 27.44% [21.43–33.45%], university graduates: 35.03% [29.73–40.34%], *p* = 0.020). The ratios by concentration of Hg exposure through fish and shellfish were 27.85% [22.10–33.60%] for the low tertile, 35.86% [29.71–42.02%] for the middle tertile, and 38.82% [32.57–45.07%] for the high tertile, indicating a clear increasing tendency with statistical significance (*p* = 0.034). The ratio of participants who surpassed the U.S. EPA threshold of 5.8 µg/L was 23.63% [20.50–26.76%] (Table 3).

## 4. Discussion

### 4.1. Relationship between Seafood Intake and Blood T-Hg Concentration

The present study explored the effectiveness of Hg exposure indices on the general management of Hg exposure, through an understanding of the relationship between Hg sources (including seafood) and the indices in women of childbearing age who are vulnerable to Hg exposure. According to the Korea National Health and Nutrition Examination Survey (KNHANES) of the Korea Centers for Disease Control and Prevention (KCDC) conducted to provide representative values for blood T-Hg in the Korean general population, the female blood T-Hg concentration by year was found to be decreasing (GM: 2.86; 2.61; 2.72; 2.25; 2.29 µg/L for the years 2007, 2008, 2009, 2010, and 2011, respectively, in female participants of all ages) [29]. The GM estimate for female participants aged 15–44 years was 3.10 µg/L [29], a lower value than 4.11 µg/L reported in the present study. It has been reported that, in Korea, those residing in regions adjacent to coasts tend to have greater seafood consumption than those in other geographical areas [30]. This might help explain the relatively higher estimation in the present study that involved participants in coastal areas, compared to data collected from participants of all geographical areas. This result might also be supported by the fact that the number of participants who surpassed the standard threshold in the present study was far higher than those in the 2010 KNHANES. Non-Korean studies on general populations have reported figures much lower relative to Korea: 0.85 µg/L (all ages, female) in the U.S. NHANES of 2009–2010 [31], 0.69 µg/L (6–79 years, female) in the Canadian Health Measures Survey (CHMS) of 2009–2011 [32], and 0.58 µg/L (18–69 years, male and female) in the relatively noncurrent German Environmental Survey Ⅲ (GerES Ⅲ) of 1998 [33]. In Japan, Nagasaki, Fukuoka, and Tokyo showed a high level of 5.18 µg/L in 2007 (19–41 years, female) [34], which might be supported by the finding that Asian populations have a higher seafood consumption rate than any other ethnic group in the world [35], and that people in island nations tend to consume more seafood than those of the interior [36,37]. It is, however, necessary to have more cumulative data before making generalizations between nations. Separately, CDC [31], Becker et al. [33], Health Canada [32], and KCDC [29] all reported that Hg exposure in men was equal to or higher than that in women, while age tended to have a positive correlation to Hg concentration. It is therefore presumed that Hg concentration in women of childbearing age would be relatively lower than in cross-gender groups or female groups from the same population. With evidence proving that low-level Hg exposure in pregnant women could negatively affect the neurological development of the fetus, it was reported that Hg concentration in cord blood might be used as a predictor of fetal neurological disorders [38], triggering widespread research on Hg concentration in cord blood and women of childbearing age and pregnancy. Referring to the 1999–2000 NHANES data, Schober et al. [39] reported that the GM of blood T-Hg was 1.02 µg/L in 1709 women aged 16–49 years, while Morrissette et al. [40] presented the equivalent figure of 0.48 µg/L in pregnant women aged 15–39 years. Results from the present study suggested that the concentration of blood T-Hg in women of childbearing age was higher than the estimates in other studies. The relationship between seafood intake and blood T-Hg concentration has been established in a few studies [41,42,43]. Among the studies with women of childbearing age as participants, Schober et al. [39] found that blood T-Hg in those who had eaten seafood in the preceding 30 days was significantly higher than others in the study group, while those who had consumed no seafood showed the lowest Hg concentration. The concentration was higher when fish was consumed, as opposed to shellfish, and highest when both fish and shellfish were consumed. The frequency of seafood consumption was also found to be closely related to the concentration. Similar to the present study, Mahaffey et al. [42] also studied women of childbearing age between 16 and 49 years, from coastal areas and noncoastal areas, to compare the results caused by expected differences in seafood diet. They found that blood T-Hg in participants from coastal areas was higher than those from noncoastal areas. People residing in coastal areas were four times more likely to surpass the threshold of 5.8 µg/L than those in noncoastal areas. In their study of pregnant women in Korea, Kim et al. [44] and Lee et al. [21] recounted that appreciation of seafood and intake frequency had positive correlations with total blood Hg. In a similar pattern to the previous literature, blood T-Hg in women of childbearing age was higher in the middle and high tertiles than in the low tertile, and the ratio of those who surpassed the 5.8 µg/L level also increased.

### 4.2. Relationship between Seafood Intake and Blood MeHg, Hair Hg, and Urine Hg Concentrations

Studies with direct analyses of blood MeHg, as opposed to other indices, are few in Korea. Lee et al. [21] reported that the AM of 59 pregnant women was 2.60 µg/L, or 85.0% of blood T-Hg, indicating a very high correlation (r = 0.937). In another study of the general population in a Korean coastal area, the GM of 400 participants was 4.05 µg/L, 78.5% of blood T-Hg [30]. The study also used the simple FFQ to estimate seafood consumption, to confirm that there was a significantly increasing tendency of T-Hg and MeHg that corresponded to seafood intake [30]. Although it might not be possible to make direct comparisons between this study and the aforementioned projects because of the differences in study period, study groups, and statistical methods applied, it is a general observation that women of childbearing age had lower Hg concentrations due to their relatively young age range. Studies with general populations showed higher estimates than the present study as they included males and the elderly. In terms of findings from non-Korean studies, Vahter et al. [45] reported that the median of blood MeHg in pregnant women was 0.94 µg/L, while Björnberg et al. [46] reported that the corresponding median value was 1.7 µg/L in 127 women aged 19–45 years, and that blood MeHg increased as their seafood intake increased. The Spearman r with the frequency of seafood intake was 0.37 (*p* < 0.001). The median value of 3.30 µg/L found in this study is much higher than values reported in non-Korean studies; however, the result that seafood intake was a main cause of increased blood MeHg was found in both the present study and the non-Korean studies. Allen et al. [47] estimated the surpass ratio of the EPA reference dose of 0.1 µg/kg/day [48] at 0.3% from hair Hg of women of childbearing aged 16–49 years in the NHANES. This was a sizeable difference compared to the ratio of 14.8% estimated in the present study. Hair Hg is a useful index to estimate long-term exposure to MeHg. The Faroe Islands cohort study is a potential representative project that looked at hair Hg in connection with seafood intake. The mean Hg concentration found in whale meat consumed in the area was 3.3 µg/g [49], while the GM of hair Hg assessed in pregnant women in the area at the time of their giving birth was 4.27 µg/g. Follow-up studies on the children born to those women showed that the GM was 1.12 µg/g when they were 12 months old, and 2.99 µg/g when they were seven years old. A neurological assessment of the children when they were seven suggested that any disorders found were correlated with Hg concentration [50]. The Seychelles Child Development Study, a cohort project that looked at the concentration of exposure to MeHg via dietary intake of fish and shellfish from deep seas and coastal areas to assess ensuing impact on health, reported that a high concentration of hair Hg might affect health status [51,52]. Spearheaded by these cohort studies, there has been dynamic research effort exploring ways to reduce MeHg concentration in susceptible and vulnerable groups. A study on hair Hg concentration in Korea found the AM of concentrations in pregnant women was 0.65 µg/g [53]. This is a relatively lower estimate than the one found in another study where the GM of residents in a few communities was 0.95 µg/g [54]. Such disparity might be due to differences in demographic characteristics and different dietary intake of seafood. Eun et al. [55] confirmed that there was an increase in hair Hg concentration that corresponded to the subjects’ preference for, and frequency of, fish consumption. Notably, a positive correlation between increasing tendency of hair Hg with frequency of eating seafood was found in a study on 111 mothers with infants under three years old. The GM of Hg concentration was 0.81 µg/g [56]. Studies conducted in several different countries confirmed that hair Hg concentrations were affected by fish and shellfish intake [57,58,59,60,61,62,63,64]. A study of women of childbearing age between 18 and 45 years reported a median of 0.32 µg/g and a linear tendency of Hg concentration increasing with the frequency of fish being eaten in a meal for a period of three months. In the cases where fish was eaten more than 20 times during the period, the concentration was seven times higher than the no-fish diet cases [65]. McDowell et al. [66] analyzed NHANES and reported that the GM of the 16–49-year age group was 0.20 µg/g and that, when fish was eaten more than three times in a 30-day period, there was a 3-fold greater concentration than cases where no fish was eaten. In their studies on pregnant women, Björnberg et al. [67] and Lindow et al. [68] confirmed that hair Hg was correlated to consumption of fish and shellfish. Compared to the results presented in this study, the concentration found in Japanese women was conspicuously higher at 1.44 µg/g [58], while the equivalent levels found in pregnant women or those of childbearing age from other countries were relatively low in general. Unlike the non-Korean studies where most of the research utilized the frequency of seafood intake, the present study looked at hair Hg concentration by the level of Hg exposure through seafood intake. Thus, direct comparisons with these studies were difficult, but correlations between seafood and hair Hg could be confirmed. There has not been much research on urine as an exposure index for I-Hg in relation to seafood consumption. The GM of urine Hg in this study was 1.72 µg/g-cr, a much higher value than that reported in the Korean National Survey of Hazardous Substances in Biological Samples Ⅱ (0.45 µg/g-cr) on the Korea general population [69]. The monitoring cohort study on exposure level and biomarkers of environmental pollutants in the industrial complex found that the GMs for each geographical region ranged 0.95–2.85 µg/g-cr, with most of the values close to 2.0 µg/g-cr [70]. This reflects that the geographical areas researched in the present study also had a high concentration of I-Hg exposure. Compared to some non-Korean studies, including NHANES (2009–2010) where the median was 0.48 µg/g-cr (all ages, female) [31] and GerES Ⅲ (1998) where the median was 0.34 µg/g-cr (18–69 years, male and female) [33], the concentration of urine Hg has been reported to be similar in the general Korean population, but the subject areas of the present study were found to have higher estimates. For the concentration of Hg exposure through fish and shellfish, the blood T-Hg was higher in the middle and high tertiles, relative to the low, but with no linear tendency observed. This might be the effect of I-Hg included in blood T-Hg. Blood MeHg, an index of O-Hg, was found to increase, corresponding to the concentration of Hg exposure through fish and shellfish. However, the concentration of urine Hg, as an index of I-Hg, was lowest in the high tertile and highest in the middle tertile. Nuttal et al. [71] also reported on the correlation of blood MeHg and urine Hg with regard to blood T-Hg. Mahaffey [72] reported that the proportion of blood MeHg increased as the T-Hg concentration rose. This was confirmed in the present study, where we observed a high correlation between blood MeHg and blood T-Hg. That there is a high ratio of blood MeHg in blood T-Hg implies that the negative impact of MeHg on health is augmented as the blood T-Hg concentration increases. It is therefore deemed necessary to provide official guidelines on seafood consumption, as in the case of the 2001 U.S. Food and Drug Administration, when the concentration of blood MeHg is observed to be high.

### 4.3. Strengths and Limitations

The present study offers important contributions in that it compared four main indices (including blood MeHg) after collecting blood, hair, and urine samples from women of childbearing age in some of the Korean coastal areas. The study also carried out a 24-h recall survey and simple FFQ to collect data for dietary intake of fish and shellfish, in order to relate Hg intake through seafood items to Hg concentration in the body. In terms of limitations, we did not observe statistically significant patterns of women of childbearing age showing blood MeHg concentration that increased in accordance with their Hg exposure through fish and shellfish, although a tendency was present. The power for the factor of Hg exposure through fish and shellfish adjusted for demographic and lifestyle characteristics was 0.24, which implies that an adequately large sample size might be needed for future follow-up studies. As for blood T-Hg, any linear relations between the biomarker and Hg exposure through fish and shellfish have not been confirmed; it might be that additional confounding factors have to be controlled for, and that a more sophisticated dietary survey has to be performed. Furthermore, the present study does not provide a data set representative of Korea, as the data were collected only from a few coastal areas in the country. Despite these drawbacks, the present study offers some contributions distinctive from other studies conducted in Korea. A large amount of data from a great number of participants were collected and analyzed to account for demographic characteristics of women of childbearing age in relation to their Hg exposure, and that blood T-Hg and blood MeHg were analyzed together to identify correlations. This study is expected to be utilized as stepping stone for follow-up research, and as basic data upon which to build relevant guidelines to help manage Hg exposure of women of childbearing age.

## 5. Conclusions

The present study found that Hg exposure through fish and shellfish in women of childbearing age can be, along with age, a main cause of increases in blood MeHg concentration. Total blood and hair Hg concentration did not show a linear relationship with Hg exposure through fish and shellfish, but the concentration was higher in the middle and high tertiles than in the low tertile. Urine Hg was not found to be related to Hg exposure through fish and shellfish, as expected. Accounting for 76.5% of blood T-Hg, blood MeHg was confirmed to be highly correlated. In the present study, a fourth of all participants were found to have a blood T-Hg concentration that was higher than the EPA standard. In addition, when compared to KNHANES, there was a disproportionately larger number of participants with high blood T-Hg concentrations in Busan, Ulsan, and Kyongnam areas, where there was greater seafood consumption relative to the rest of the country. In conclusion, it is necessary to pay attention to Hg exposure through fish and shellfish in women of childbearing age, considering the seriousness of the exposure in this particular demographic group.

## Figures and Tables

**Figure 1 ijerph-17-04907-f001:**
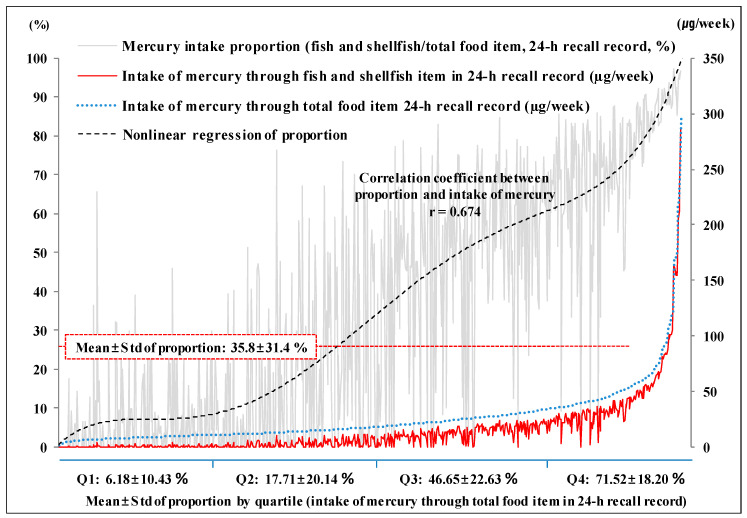
Mercury intake proportion from fish and shellfish items in a 24-h recall record. Q1–Q4: categorization by quartile according to the total mercury intake; Q1: 1.285–10.935 µg/week, Q2: 10.939–17.490 µg/week, Q3: 17–528–30.931 µg/week, Q4: 31.024–295.803 µg/week.

**Figure 2 ijerph-17-04907-f002:**
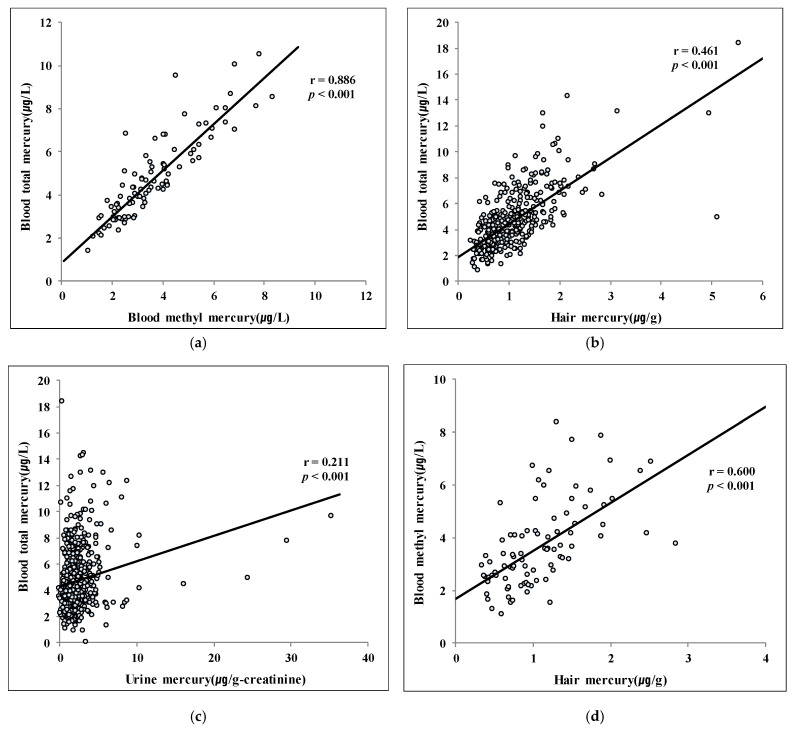
Correlations among mercury concentrations (**a**) Total blood mercury and blood methyl mercury; (**b**) Total blood mercury and hair mercury; (**c**) Total blood mercury and urine mercury; (**d**) Blood methyl mercury and hair mercury.

**Table 1 ijerph-17-04907-t001:** Geometric means of the total blood, blood methyl, hair, and urine mercury concentration.

Factor	Geometric Mean (95% Confidence Interval)											
	Blood Total Mercury (µg/L)			Blood Methyl Mercury (µg/L)			Hair Mercury (µg/g)			Urine Mercury (µg/g-creatinine) ^b^		
	*n*	Crude	Adjusted ^c^	*n*	Crude	Adjusted ^c^	*n*	Crude	Adjusted ^c^	*n*	Crude	Adjusted ^c^
Total	711	4.11 (3.90–4.33)		90	3.32 (3.03–3.64)		359	0.97 (0.92–1.03)		616	1.72 (1.61–1.85)	
Age (y)												
19–29	233	3.69 (3.49–3.90)	3.62 (3.21–4.08)	30	2.96 (2.63–3.33)	2.77 (2.24–3.44)	114	0.84 (0.77–0.93)	0.79 (0.70–0.89)	191	1.52 (1.36–1.70)	1.45 (1.24–1.71)
30–39	264	4.08 (3.61–4.60)	3.85 (3.49–4.24)	30	3.59 (3.06–4.20)	3.55 (2.93–4.30)	123	1.06 (0.97–1.16)	1.00 (0.90–1.11)	237	1.77 (1.54–2.04)	1.74 (1.53–1.98)
40–49	214	4.65 (4.36–4.96)	4.32 (3.87–4.83)	30	3.43 (2.81–4.20)	3.49 (2.87–4.25)	122	1.02 (0.94–1.12)	0.97 (0.87–1.08)	188	1.90 (1.73–2.09)	1.90 (1.65–2.19)
*p*-value		0.003	0.059		0.206	0.118		0.001	0.002		0.042	0.028
Education												
High School or Less	182	4.51 (4.20–4.84)	4.21 (3.72–4.75)	19	3.14 (2.53–3.90)	3.08 (2.45–3.88)	91	1.02 (0.92–1.12)	0.93 (0.82–1.05)	162	1.74 (1.43–2.13)	1.60 (1.37–1.88)
University Student	215	3.47 (3.00–4.01)	3.37 (3.01–3.77)	26	3.49 (2.93–4.16)	3.41 (2.74–4.24)	107	0.89 (0.79–1.00)	0.86 (0.77–0.96)	184	1.73 (1.58–1.91)	1.74 (1.50–2.03)
University Graduate	314	4.36 (4.16–4.58)	4.25 (3.86–4.67)	45	3.30 (2.89–3.76)	3.27 (2.76–3.88)	161	1.01 (0.94–1.08)	0.96 (0.87–1.05)	270	1.71 (1.56–1.86)	1.72 (1.52–1.95)
*p*-value		<0.001	0.001		0.722	0.791		0.102	0.229		0.963	0.669
Income (KRW 10,000)												
≤199	186	4.09 (3.85–4.35)	3.93 (3.50–4.40)	20	3.30 (2.70–4.03)	3.52 (2.81–4.40)	90	0.98 (0.90–1.08)	0.94 (0.84–1.05)	159	1.97 (1.76–2.21)	1.98 (1.71–2.30)
200–299	147	4.11 (3.78–4.47)	3.94 (3.48–4.47)	16	2.78 (2.18–3.54)	2.72 (2.11–3.50)	69	0.99 (0.87–1.12)	0.91 (0.80–1.04)	123	1.72 (1.52–1.93)	1.65 (1.40–1.95)
300–399	150	4.37 (4.05–4.72)	4.06 (3.58–4.60)	21	3.53 (2.92–4.26)	3.39 (2.72–4.22)	75	0.93 (0.82–1.05)	0.87 (0.77–0.99)	132	1.49 (1.19–1.88)	1.43 (1.21–1.68)
≥400	190	4.04 (3.43–4.76)	3.75 (3.33–4.23)	28	3.50 (2.92–4.18)	3.45 (2.83–4.19)	107	1.01 (0.91–1.12)	0.93 (0.83–1.05)	167	1.81 (1.63–2.01)	1.73 (1.48–2.03)
*p*-value		0.781	0.795		0.339	0.324		0.774	0.794		0.057	0.018
Drinking Status												
Current Drinker	443	4.11 (3.80–4.44)	4.08 (3.80–4.39)	53	3.19 (2.82–3.60)	3.19 (2.79–3.64)	223	0.99 (0.92–1.06)	0.96 (0.90–1.04)	379	1.75 (1.59–1.94)	1.78 (1.62–1.96)
Past Drinker	69	3.75 (3.37–4.17)	3.62 (3.04–4.32)	10	3.54 (2.70–4.64)	3.24 (2.36–4.43)	35	0.90 (0.77–1.06)	0.85 (0.71–1.01)	63	1.79 (1.48–2.17)	1.71 (1.36–2.15)
Non Drinker	199	4.24 (3.97–4.53)	4.07 (3.67–4.52)	27	3.50 (2.92–4.20)	3.34 (2.75–4.05)	101	0.98 (0.88–1.08)	0.94 (0.84–1.04)	174	1.64 (1.48–1.81)	1.58 (1.38–1.82)
*p*-value		0.466	0.451		0.594	0.926		0.640	0.390		0.666	0.378
Amalgam Treatment												
Absence	272	3.82 (3.38–4.30)	3.63 (3.29–4.02)	37	3.32 (2.87–3.83)	3.22 (2.69–3.86)	144	0.90 (0.83–0.97)	0.84 (0.76–0.93)	238	1.72 (1.57–1.88)	1.68 (1.47–1.91)
Presence	439	4.30 (4.12–4.48)	4.23 (3.88–4.60)	53	3.32 (2.94–3.75)	3.28 (2.81–3.84)	215	1.03 (0.96–1.11)	0.99 (0.91–1.08)	378	1.73 (1.56–1.91)	1.70 (1.52–1.90)
*p*-value		0.031	0.009		0.999	0.848		0.014	0.004		0.944	0.882
Mercury Exposure through Fish and Shellfish ^a^												
Low (≤ 663 µg/y)	237	3.84 (3.62–4.07)	3.68 (3.32–4.09)	26	3.20 (2.70–3.80)	3.06 (2.48–3.77)	119	0.94 (0.85–1.03)	0.89 (0.80–0.99)	201	1.67 (1.50–1.85)	1.67 (1.45–1.93)
Middle (664–1294 µg/y)	237	4.31 (4.07–4.55)	4.08 (3.67–4.53)	37	3.21 (2.76–3.74)	3.12 (2.59–3.76)	121	1.00 (0.91–1.10)	0.93 (0.84–1.03)	208	1.92 (1.74–2.11)	1.86 (1.63–2.14)
High (≥ 1295 µg/y)	237	4.19 (3.66–4.80)	4.01 (3.59–4.47)	27	3.58 (3.02–4.25)	3.60 (2.95–4.40)	119	0.99 (0.90–1.08)	0.92 (0.83–1.03)	207	1.60 (1.37–1.87)	1.54 (1.34–1.78)
*p*-value		0.183	0.281		0.553	0.394		0.583	0.739		0.553	0.101

^a^: Simple 22-item Food Frequency Questionnaire ^b^: Excluded 30 mg/dL < creatinine or 300 mg/dL > creatinine (world health organization standard) ^c^: Adjusted for all demographic and lifestyle variables in the table.

**Table 2 ijerph-17-04907-t002:** Proportion of blood MeHg/T-Hg concentrations.

Factor	Arithmetic Means of Proportion (95% Confidence Interval)		
	*n*	Crude	Adjusted ^b^
Total	90	76.51 (73.62–79.39)	
Age (y)			
19–29	30	77.35 (72.24–82.46)	78.67 (72.19–85.16)
30–39	30	78.07 (72.45–83.69)	79.58 (73.79–85.37)
40–49	30	74.10 (69.36–78.84)	76.82 (70.88–82.75)
*p*-value		0.499	0.771
Education (y)			
High School or Less	19	74.34 (68.58–80.10)	78.05 (71.08–85.01)
University Student	26	80.40 (74.26–86.55)	81.88 (75.30–88.47)
University Graduate	45	75.17 (71.11–79.22)	75.14 (69.96–80.32)
*p*-value		0.229	0.217
Income (KRW 10,000)			
≤199	20	77.52 (70.50–84.55)	80.21 (73.44–86.97)
200–299	16	72.06 (65.76–78.36)	74.81 (67.19–82.43)
300–399	21	75.54 (68.38–82.70)	77.15 (70.53–83.77)
≥400	28	78.99 (74.37–83.61)	81.26 (75.34–87.19)
*p*-value		0.425	0.457
Drinking Status			
Current Drinker	53	74.61 (70.76–78.47)	72.95 (68.92–76.99)
Past Drinker	10	83.13 (74.42–91.83)	82.10 (72.62–91.58)
Non Drinker	27	77.77 (72.41–83.12)	80.02 (74.15–85.89)
*p*-value		0.173	0.051
Amalgam Treatment			
Absence	37	73.92 (69.51–78.33)	76.87 (71.38–82.36)
Presence	53	78.31 (74.45–82.17)	79.84 (75.11–84.58)
*p*-value		0.138	0.348
Mercury Exposure through Fish and Shellfish ^a^			
Low (≤ 663 µg/y)	26	75.70 (70.88–80.52)	76.30 (69.96–82.63)
Middle (664–1294 µg/y)	37	75.64 (69.77–81.51)	77.54 (71.53–83.55)
High (≥ 1295 µg/y)	27	77.70 (72.87–82.54)	81.24 (75.56–86.91)
*p*-value		0.793	0.386

^a^: Simple 22-item Food Frequency Questionnaire ^b^: Adjusted for all demographic and lifestyle variables in the table.

**Table 3 ijerph-17-04907-t003:** Proportions of total blood mercury concentrations exceeding the standard criteria.

Factor	Proportion (95% Confidence Interval)		
	*n*	≥5.0 (HBM-I)	≥5.8 (EPA)
Total	711	34.18 (30.68–37.67)	23.63 (20.50–26.76)
Age (y)			
19–29	233	21.03 (15.76–26.30)	14.16 (9.65–18.67)
30–39	264	37.50 (31.62–43.38)	26.52 (21.16–31.87)
40–49	214	44.39 (37.68–51.10)	30.37 (24.16–36.58)
*p*-value		<0.001	<0.001
Education (y)			
High School or Less	182	40.66 (33.46–47.86)	28.57 (21.95–35.20)
University Student	215	27.44 (21.43–33.45)	17.21 (12.12–22.30)
University Graduate	314	35.03 (29.73–40.34)	25.16 (20.33–29.99)
*p*-value		0.020	0.020
Income (KRW 10,000)			
≤199	186	29.03 (22.45–35.62)	18.28 (12.67–23.89)
200–299	147	34.69 (26.91–42.48)	27.21 (19.93–34.49)
300–399	150	38.67 (30.78–46.55)	28.00 (20.73–35.27)
≥400	190	38.95 (31.95–45.94)	25.79 (19.51–32.07)
*p*-value		0.167	0.131
Drinking Status			
Current Drinker	443	34.54 (30.09–38.98)	23.93 (19.94–27.92)
Past Drinker	69	21.74 (11.76–31.72)	11.59 (3.85–19.34)
Non Drinker	199	37.69 (30.90–44.48)	27.14 (20.90–33.37)
*p*-value		0.053	0.031
Amalgam Treatment			
Absence	272	30.88 (25.36–36.41)	20.96 (16.09–25.82)
Presence	439	36.22 (31.71–40.73)	25.28 (21.20–29.37)
*p*-value		0.145	0.187
Mercury Exposure through Fish and Shellfish ^a^			
Low (≤ 663 µg/y)	237	27.85 (22.10–33.60)	17.72 (12.82–22.62)
Middle (664–1294 µg/y)	237	35.86 (29.71–42.02)	25.74 (20.13–31.34)
High (≥ 1295 µg/y)	237	38.82 (32.57–45.07)	27.43 (21.70–33.15)
*p*-value		0.034	0.029

^a^: Simple 22-item Food Frequency Questionnaire; HBM: German Human Biomonitoring Commission; EPA: United States Environmental Protection Agency.

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
