# Peer review of "The Relationship between Mercury Exposure Indices and Dietary Intake of Fish and Shellfish in Women of Childbearing Age"

_ijerph, 2020, doi:10.3390/ijerph17134907_

Round 1

Reviewer 1 Report

Comment 1: The introduction should include more information regarding  fish and shellfish of concern in Busan, Ulsan and Kyongham with MeHg levels found in fish from those areas.

Comment 2: In methods, study design and data collection: Why was only 90 out of 711 samples were analyzed for MeHg in blood?

Comment 3: 2.2 Questionnaire survey: Simple FFQ estimated the amount of Hg intake that occurred from consuming 22 different seafood items including 20 regular fish and shellfish; however the results indicate 16 food groups please correct methods and/or results.

Comment 4: 2.3 Analyses of blood, hair, and urine mercury procedures should be described separately. 2.3.1 Blood, 2.3.2 hair etc.

Comment 5: If contamination was identified, what were consequences? On line 100, it just states checked?

Results

3.1 Mercury intake through fish and shellfish

Comment 6: Was the Hg content in 16 food groups known?

Comment 7: Did authors consider testing the fish or shellfish consumed by participants in the study?

Comment 8: The conclusion is a great leap without sufficient data to suggest Hg exposure through fish and shellfish in woman of childbearing age was a main cause of increases in blood MeHg concentrations. Conclusions should be revised to provide a better rationale for results obtained.

Reviewer 2 Report

In this study, the authors investigated the relationship between mercury intake through fish and shellfish consumption and mercury exposure indices in 711 women of childbearing age from coastal areas in Korea. The results showed that mercury exposure through fish and shellfish intake is a main factor for an increase of blood methyl mercury concentration in women of childbearing age. This suggests strongly that more attention needs to be paid to mercury exposure through seafood intake in women of childbearing age. Korea has failed to provide guidelines and care for children and women of childbearing age who are at particularly high risk. Hence, I think that data from this study are valuable for standards and guidelines to be established in Korea.

However, readers may have much difficulty in reading this paper, because the paper is not divided properly into paragraphs. In particular, the discussion section has many problems. It should be logically reorganized and divided into some small sections, such as “4.1. Relationship between seafood intake and blood T-Hg concentration” and “4.2. Relationship between seafood intake and blood MeHg concentration”. Moreover, part on lines 155–187 will be suitable for summary or conclusion. This also needs to reconsider the conclusion section. Oher minor comments are given below.

Lines 15–16: This sentence should be revised to “711 women of childbearing age from coastal areas with a high concentration of mercury exposure in Korea were studied”.

Lines 15–16: What is the reason why the concentration is not higher when shellfish was consumed?

Lines 143–144: The authors mentioned that a remarkable difference was observed in the shellfish intake frequency. Where is this result shown?

Lines 200–202: This sentence means the same as the sentence on lines 195–196. Avoid a repetition.

Round 2

Reviewer 1 Report

Comments were addressed

Reviewer 2 Report

The revised manuscript has been satisfactorily improved. Hence, this is acceptable for publication in the present form.